# Effect of Autotransfusion in HCC Surgery on Survival and Recurrence: A Systematic Review and Meta-Analysis

**DOI:** 10.3390/cancers14194837

**Published:** 2022-10-03

**Authors:** Anastasia Murtha-Lemekhova, Juri Fuchs, Emil Ritscher, Katrin Hoffmann

**Affiliations:** Department of General, Visceral and Transplantation Surgery, Heidelberg University Hospital, 69120 Heidelberg, Germany

**Keywords:** autotransfusion, blood salvage, HCC, hepatectomy, liver transplantation, meta-analysis, survival, recurrence

## Abstract

**Simple Summary:**

Administering patients their own blood during liver surgery would reduce the burden on blood banks and immunologic reactions to foreign blood products. Two methods of autotransfusion are available: scheduled donation before surgery and salvage during surgery (intraoperative blood salvage, IBS). However, concerns over circulating tumor cells dissuade against autotransfusion in patients undergoing liver surgery for hepatocellular carcinoma (HCC). This meta-analysis evaluated available reports on autotransfusion, including the reintroduction of blood collected from the surgical area during a tumor operation. Patients who received blood collected from the surgical site during liver transplantation did not develop more recurrences of HCC and their overall survival was similar to patients who received donor blood products. Patients undergoing liver resection mostly received blood they donated prior to surgery. They showed a better overall survival as well as cancer-free survival after surgery. Randomized controlled trials are needed to better estimate the effects of autotransfusion on patients and studies incorporating autotransfusion of blood collected during liver resection are needed.

**Abstract:**

Background: The chronic blood shortage has forced clinicians to seek alternatives to allogeneic blood transfusions during surgery. Due to anatomic uniqueness resulting in a vast vasculature, liver surgery can lead to significant blood loss, and an estimated 30% of patients require blood transfusions in major hepatectomy. Allogeneic transfusion harbors the risk of an immunologic reaction. However, the hesitation to reinfuse a patient’s own blood during cancer surgery is reinforced by the potentiality of reintroducing and disseminating tumor cells into an individual undergoing curative treatment. Two methods of autotransfusions are common: autotransfusion after preoperative blood donation and intraoperative blood salvage (IBS). We aim to investigate the effect of autotransfusion on recurrence and survival rates of patients undergoing surgery for HCC. Methods: The protocol for this meta-analysis was registered at PROSPERO prior to data extraction. MEDLINE, Web of Science and Cochrane Library were searched for publications on liver surgery and blood salvage (autologous transfusion or intraoperative blood salvage). Comparative studies were included. Outcomes focused on long-term oncologic status and mortality. Hazard ratios (HR) estimated outcomes with a fixed-effects model. Risk of bias was assessed using ROBINS-I, and certainty of evidence was evaluated with GRADE. Separate analyses were performed for liver transplantation and hepatectomies. Results: Fifteen studies were included in the analysis (nine on transplantation and six on hepatectomies), and they comprised 2052 patients. Overall survival was comparable between patients who received intraoperative blood salvage (IBS) or not for liver transplantation (HR 1.13, 95% CI [0.89, 1.42] *p* = 0.31). Disease-free survival also was comparable (HR 0.97, 95% CI [0.76, 1.24], *p* = 0.83). Autotransfusion after prior donation was predominantly used in hepatectomy. Patients who received autotransfusion had a significantly better overall survival than the control (HR 0.71, 95% CI [0.58, 0.88], *p* = 0.002). Disease-free survival was also significantly higher in patients with autotransfusion (HR 0.88, 95% CI [0.80, 0.96], *p* = 0.005). Although overall, the certainty of evidence is low and included studies exhibited methodological heterogeneity, the heterogeneity of outcomes was low to moderate. Conclusion: Autotransfusion, including intraoperative blood salvage, does not adversely affect the overall or disease-free survival of patients with HCC undergoing resection or transplantation. The results of this meta-analysis justify a randomized-controlled trial regarding the feasibility and potential benefits of autotransfusion in HCC surgery.

## 1. Introduction

The liver has a myriad of functions in the human body, and is therefore central for survival. As the liver is the main metabolizer of noxa, drugs, and metabolites, and due to its strategic blood supply, it is unsurprising that the liver is frequently afflicted by cancer. Due to alcohol, hepatitis, and metabolic syndrome, the number of hepatocellular carcinoma (HCC) diagnoses is on the rise [1]. HCC is the main indicator for liver transplantation, along with chronic diseases that facilitate HCC development [2]. As a curative treatment or bridging therapy, HCC resection is offered globally to approximately 150,000 patients annually [3].

However, the anatomic peculiarities of the liver mean that liver surgery is associated with frequent significant blood loss, and about 30% of patients require transfusions during major hepatectomies [4]. Liver transplantation is often associated with blood loss of 2–3 L, whereas liver resection has a wide span depending on the area of parenchymal transection, intracavital pressures and techniques used [5]. As hepatobiliary surgery is expanding its arsenal and more patients are offered a curative surgical approach for their cancers, intraoperative requirements are also growing in importance. Blood transfusions are often necessary during hepatobiliary surgery, and few transplantations are completed without it. Allogenic blood products should never be administered heedlessly due to potential immunologic reactions and other associated complications [6]. Recent depletions of blood products necessitating the suspension of elective procedures underlined the need for evidence-based evolution in this area of surgery as well [7]. In addition to preoperative autologous donation of blood products which can then be used intraoperatively, the option of intraoperative blood salvage through aspiration, filtering, and re-administration may reduce the need for allogenic transfusions [8]. Although the infusion of autologous blood bears miniscule risk of immunologic reactions, surgeons are cautious regarding it due to a potential risk of reintroducing and disseminating tumor cells into the bloodstream, therefore aiding development of metastasis [9]. This is particularly scrutinized in continuous autotransfusion, e.g., IBS. The aim of this meta-analysis was to compare how auto-transfusion affects recurrence and survival rates in patients with HCC after liver surgery.

## 2. Materials and Methods

Current PRISMA guidelines served as the reporting reference [10], as well as the Cochrane Handbook for Systematic Reviews and Interventions [11]. The meta-analysis was registered at PROSPERO prior to data extraction (CRD42022352343). In accordance with the recent recommendations [12], PubMed, Web of Science and Cochrane Library were searched for publications on autotransfusion in hepatobiliary surgery. The search strategy focused on liver transplantation and hepatectomy (Appendix A) and was completed on 24 June 2022. No restrictions for language or publication year were implemented. A hand search through the references of the included studies was performed to identify additional manuscripts.

PICOS criteria utilized:Population: patients undergoing liver surgery for HCCIntervention: autologous transfusion (including intraoperative blood salvage)Comparison: no autologous transfusionOutcomes: recurrence, disease-free survival and overall survivalStudies: comparative studies irrespective on methodology

Two independent reviewers screened titles and abstracts first and then full texts (AML and JF), consulting a third reviewer (KH) in cases of disagreements. After agreement on the studies to include, the same reviewers (AML and JF) extracted pertinent information into a standardized form containing following domains: publication information (title, authors, year of publication, journal, country of data origin, funding), methodology (design and ROBINS-I domains), and clinical data (cohorts’ characteristics, interventions, outcomes of interests). Variables were pooled if described in at least two reports. Estimated effects of survival analysis were extracted and analyzed as hazard ratios [13]. For dichotomous effects, odds ratios pooled with the Mantel–Haenszel method were used. Meta-analyses were performed with Review Manager version 5.3 (The Cochrane Collaboration, Oxford, UK) with forest plots depicting effect estimates. A fixed-effects model was utilized for all outcomes. Statistical heterogeneity was evaluated using the I2 statistics with 25% indicating the threshold from low to moderate and 75% as the threshold to high heterogeneity. The methodological quality of included studies was performed using ROBINS-I [14]. The certainty of evidence was assessed using GRADE [15,16].

## 3. Results

The study selection is depicted in the PRISMA flow diagram (Figure 1).

In total, fifteen studies analyzing 2052 patients were included in this meta-analysis: nine for intraoperative blood salvage during liver transplantation for HCC and six for autotransfusion during hepatectomy for HCC [5,8,9,17,18,19,20,21,22,23,24,25,26,27,28]. Two studies reported on the same cohort, with additional follow-up provided in the second study, and so are referred to as a merged report [20,21]. The characteristics of the included studies are provided in Table 1.

### 3.1. Autotransfusion in Liver Transplantation

All studies on autotransfusion in liver transplantation involved intraoperative blood salvage and, with the exception of one study, were retrospective in nature. Table 2 provides aggregated characteristics and significance testing.

From eight studies, data on disease-free survival could be extracted and the overall effect expressed as the hazard ratio. The overall effects were similar in both groups, where IBS was used and the control (HR 0.98, 95% CI [0.76,1.24], *p* = 0.83), with studies showing moderate heterogeneity (Figure 2).

Data on recurrence were provided by six studies. The pooled odds ratio for recurrence was similar in both groups (OR 0.71 95% CI [0.41, 1.23], *p* = 0.22). The heterogeneity was low (Figure 3).

Six studies contributed data on overall survival, with the pooled overall effect being similar in patients with IBS and without (HR 1.13. 95% CI [0.89, 1.42], *p* = 0.31) (Figure 4). The studies showed low heterogeneity.

### 3.2. Autotransfusion in Hepatectomy

Six studies reported on autotransfusion in hepatectomy for HCC, with two studies reporting on the same cohort. One study reported on IBS use, while all others utilized preoperative phlebotomy and autotransfusion. In two studies, rh-EPO was used on an individual basis after phlebotomy. Table 3 provides a summary of the characteristics pooled from the studies.

Based on disease-free survival data provided by four reports, an overall hazard ratio was calculated, which signified that patients who received autotransfusion had a significantly better DFS compared to the control (HR 0.88 95% CI [0.80, 0.96], *p* = 0.005) (Figure 5).

Two studies provided raw data on recurrence, with a pooled effect lacking a significant difference between patients that received autotransfusion and those who did not (OR 0.28, 95% CI [0.04, 2.18], *p* = 0.22) (Figure 6).

Overall survival was extracted from three reports. Patients who received autotransfusions had a significantly better overall survival compared to the control (HR 0.71, 95% CI [0.58, 0.88], *p* = 0.002) (Figure 7).

### 3.3. Risk of Bias and Certainty of Evidence

Overall, studies presented a low to moderate risk of bias. Most studies presented a low risk of bias, although some had a moderate overall risk of bias, largely due to potential confounding or selection bias (Table 4).

The certainty of evidence was assessed using GRADE, and the assessed outcomes are presented in Table 5. Due to the predominantly retrospective study methodology, the certainty of evidence ranges from very low to low.

## 4. Discussion

Autotransfusion in hepatobiliary surgery is scarcely utilized for tumor surgeries due to concerns over the inadvertent reintroduction of circulating tumor cells (CTCs) into the bloodstream, effectively causing metastasis. Blood collected from the surgical field during oncological surgery may contain tumor cells, and thus the potentiality is there. There are indications that as the liver is mobilized during surgery, the tumor cells are dislodged and disseminated [29]. The technique used during hepatectomy may influence the dissemination, and the anterior approach, with parenchymal transection and venous control before right lobe mobilization, may limit it [29,30]. The mechanisms and real risk of seeding metastasis from CTCs is unclear in HCC. For instance, needle track seeding after percutaneous procedures (fine needle aspiration biopsy, percutaneous ethanol injection and percutaneous transhepatic biliary drainage) developed in 0.16% in the analysis of 50,920 patients [31]. In hepatobiliary surgery, the no-touch technique during oncologic surgery is the uncompromising standard to prevent extrahepatic seeding. However, high levels of circulating tumor cells (over 3.5 cells per 10 mL of blood) are associated with a higher recurrence rate after hepatectomy [29].

Based on the evidence synthesized in this meta-analysis, IBS in liver transplantation for HCC should be considered, as studies have not shown a negative effect on survival or disease progression. However, the certainty of evidence is low, predominantly due to the retrospective methodology of most studies. Thus, caution should be exerted before the introduction of IBS into clinical practice; however, a randomized-controlled trial is justified and long overdue. In hepatic resection, IBS has only been evaluated in one study [20], which did not show a clinical disadvantage for patients. In conjunction with liver transplantation studies, the evidence supports further evaluation of IBS in hepatic surgery in a controlled trial. Penultimately, a study investigating tumor cell presence in blood aspirated from the surgical field may provide additional evidence to structure the trial with adequate safeguards in place.

Autologous blood transfusion after preoperative donation appears to be less controversial; however, not all patients may qualify for this intervention, as tumor anemia is highly prevalent in HCC patients [32]. As most studies on autotransfusion after preoperative donation are non-randomized retrospective analyses, a bias in the selection of patients must be anticipated, which may explain the significantly better outcomes in patients after autotransfusion. As shown in the comparison of aggregated baseline characteristics, significantly more patients in the group who received autotransfusion before hepatectomy had Child–Pugh A class, which may influence survival.

A limitation of the meta-analysis that should further be addressed in randomized trials is the comparability of compared groups. Most variables were not reported by all studies, and thus aggregate data also bear potential bias. In both comparisons, significant differences were detected in the aggregate data of both groups, concerning preoperative patient criteria. These need to be considered in the structure of further studies. Additionally, some studies failed to report detailed survival data, and thus time-to-event data had to be inferred [13]. Overall, the number of patients is relatively small in the comparisons for autotransfusion and IBS in liver transplantation and hepatectomy. The reports were observational and largely retrospective. Hence, the certainty of evidence for outcomes analyzed in this meta-analysis is very low, and further randomized trials are needed.

## 5. Conclusions

Intraoperative blood salvage during liver transplantation for HCC does not lead to poorer disease-free or overall survival, but must be evaluated further in a randomized-controlled trial. Intraoperative blood salvage in hepatectomy for HCC is under-evaluated and trials incorporating the quantification of tumor cells in the aspirate from the surgical field should shed more light on recurrence risk patterns associated with it.

## Figures and Tables

**Figure 1 cancers-14-04837-f001:**
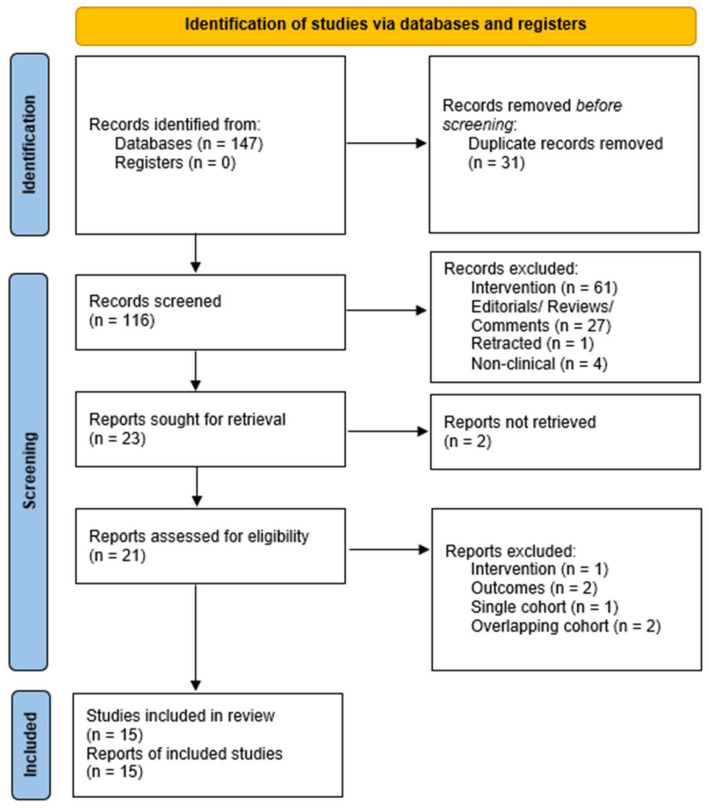
Study selection process.

**Figure 2 cancers-14-04837-f002:**
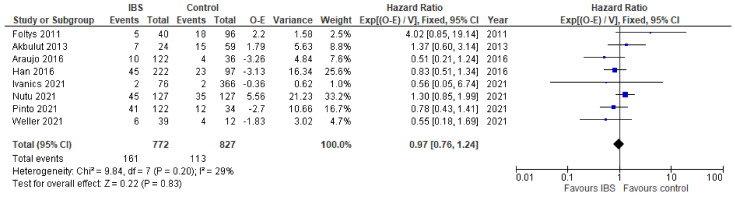
Forest plot for disease-free survival in liver transplant patients with IBS and without. Blue squares indicate individual effects and black diamond illustrates overall effect.

**Figure 3 cancers-14-04837-f003:**
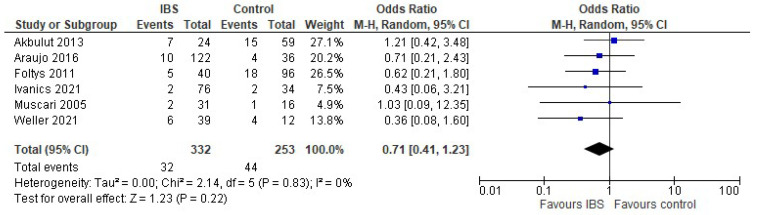
Forest plot for recurrence in liver transplant patients with IBS and without. Blue squares indicate individual effects and black diamond illustrates overall effect.

**Figure 4 cancers-14-04837-f004:**
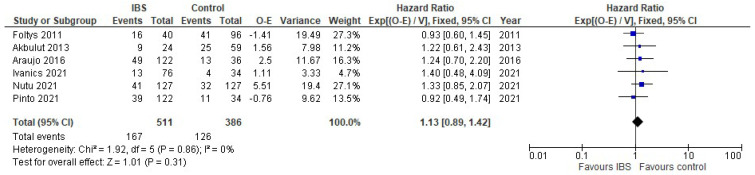
Forest plot for overall survival in liver transplant patients with IBS and without. Blue squares indicate individual effects and black diamond illustrates overall effect.

**Figure 5 cancers-14-04837-f005:**
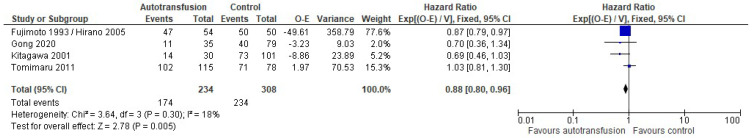
Forest plot for disease-free survival in hepatectomy patients with autotransfusion and without. Blue squares indicate individual effects and black diamond illustrates overall effect.

**Figure 6 cancers-14-04837-f006:**
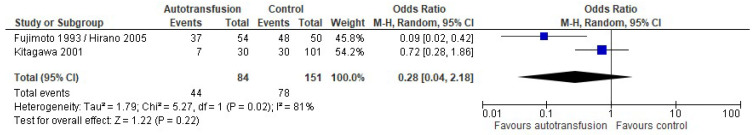
Forest plot for recurrence in hepatectomy patients with autotransfusion and without. Blue squares indicate individual effects and black diamond illustrates overall effect.

**Figure 7 cancers-14-04837-f007:**
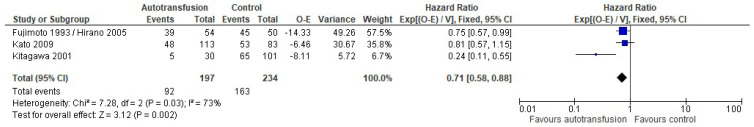
Forest plot for overall survival in hepatectomy patients with autotransfusion and without. Blue squares indicate individual effects and black diamond illustrates overall effect.

**Table 1 cancers-14-04837-t001:** Overview of included studies.

Study	Country	Population	Design	Intervention	Control
Akbulut 2013 [17]	Turkey	Liver transplantation Living and deceased donors	Retrospective	IBS	no IBS
Araujo 2016 [18]	Brazil	Liver transplantation	Retrospective	IBS	no IBS
Foltys 2011 [19]	Germany	Liver transplantation	Retrospective	IBS	no IBS
Han 2016 [8]	Korea	Liver transplantation Living donors	Retrospective, propensity score matched	IBS, leucocyte depletion	no IBS
Ivanics 2021 [22]	Canada	Liver transplantation Living and deceased donors	Retrospective	IBS	no IBS
Muscari 2005 [25]	France	Liver transplantation Deceased donor	Prospective	IBS	no IBS
Nutu 2021 [26]	UK	Liver transplantation Deceased donor	Retrospective, propensity score matched	IBS	no IBS
Pinto 2021 [5]	Brazil	Liver transplantation Deceased donor	Retrospective	IBS	no IBS
Weller 2021 [28]	Germany	Liver transplantation	Retrospective	IBS ± irradiation	no IBS
Fujimoto 1993/Hirano 2005 [20,21]	Japan	Hepatectomy	Prospective	Autotransfusion after preoperative phlebotomy + IBS	no IBS
Gong 2020 [9]	China	Hepatectomy	Prospective	Autotransfusion after preoperative phlebotomy	Allogeneic transfusions
Kato 2009 [23]	Japan	Hepatectomy	Prospective	Autotransfusion after preoperative phlebotomy + rh-EPO	no autotransfusion
Kitagawa 2001 [24]	Japan	Hepatectomy	Prospective	Autotransfusion after preoperative phlebotomy	no autotransfusion (allogeneic or no transfusion)
Tomimaru 2011 [27]	Japan	Hepatectomy	Prospective	Autotransfusion after preoperative phlebotomy + rh-EPO	no transfusion

IBS: intraoperative blood salvage. rh-EPO: recombinant human EPO.

**Table 2 cancers-14-04837-t002:** Aggregated characteristics summary for liver transplantation patients.

Characteristic	IBS (*n* = 803)	no IBS (*n* = 511)	Significance (*p* =)
Age (years) [mean ± SD]	55.8 ± 6.6	55.7 ± 6.2	0.82
Gender (m/f)	535/141	311/71	0.38
BMI [mean ± SD]	28.0 ± 4.25	27.3 ± 3.67	0.02
- Liver disease			0.08
- Alcohol-associated	70	70	
- Metabolic-associated	30	14	
- Hepatitis	326	221	
- Genetic	2	3	
- Child-Pugh-Turcott			<0.001
- A	37	78	
- B	50	70	
- C	84	56	
MELD [mean ± SD]	13.1 ± 5.3	12.6 ± 3.9	0.31
HCC diameter	3.8 ± 1.5	4.1 ± 1.5	0.07
- Grading			0.47
- Well differentiated	50	58	
- Moderately differentiated	134	123	
- Poorly differentiated	16	20	
			Vascular invasion
- Donor			0.68
- Diseased donors	338	264	
- Living donors	201	166	

**Table 3 cancers-14-04837-t003:** Aggregated characteristics summary for hepatectomy patients.

Characteristic	Autotransfusion (*n* = 347)	No Autotransfusion (*n* = 391)	Significance
Age (years) [mean ± SD]	58.7 ± 5.5	57.8 ± 4.1	0.01
Gender (m/f)	234/48	167/44	0.28
- Cirrhosis			0.08
- Present	88	176	
- Absent	144	215	
Child-Pugh-Turcott			<0.001
- A	199	142	
- B/C	64	97	
HCC tumor			0.14
- Solitary	154	120	
- Multiple	74	41	
Vascular invasion	134	132	0.36
Intraoperative blood loss	1212 ± 998	2056 ± 2123	<0.001

**Table 4 cancers-14-04837-t004:** Risk of bias of included studies.

	Bias Due to Confounding	Bias in Selection of Participants into the Study	Bias in Classification of Interventions	Bias Due to Deviations from Intended Interventions	Bias Due to Missing Data	Bias in Measurement of Outcomes	Bias in Selection of the Reported Results	Overall
Akbulut 2013 [17]	!	?	+	+	?	?	+	+
Araujo 2016 [18]	+	?	+	+	+	+	+	+
Foltys 2011 [19]	!	!	+	?	+	+	+	?
Han 2016 [8]	+	?	+	+	+	+	+	+
Ivanics 2021 [22]	+	!	+	+	+	+	+	+
Muscari 2005 [25]	?	?	+	+	?	?	+	?
Nutu 2021 [26]	+	+	+	?	+	+	+	+
Pinto 2021 [5]	?	?	+	+	+	+	+	+
Weller 2021 [28]	!	!	+	+	?	?	+	?
Fujimoto 1993/Hirano 2005 [20,21]	+	+	+	+	?	+	?	+
Gong 2020 [9]	+	+	+	+	?	+	?	+
Kato 2009 [23]	+	+	+	+	+	+	+	+
Kitagawa 2001 [24]	+	+	+	+	?	?	+	+
Tomimaru 2011 [27]	+	+	+	+	+	+	+	+

+ low risk of bias; ? moderate risk of bias; ! high risk of bias.

**Table 5 cancers-14-04837-t005:** Certainty of evidence for assessed outcomes.

Outcome	№ of Included Studies	Certainty of the Evidence (GRADE)	Relative Effect (95% CI)
DFS after IBS vs. no IBS in LTx	8	Very Low	HR 0.98 [0.76,1.24]
Recurrence after IBS vs. no IBS in LTx	6	Very Low	OR 0.71 CI [0.41, 1.23]
OS after IBS vs. no IBS in LTx	6	Very Low	HR 1.13. [0.89, 1.42]
DFS after autotransfusion vs. none in hepatectomy	4	Very Low	HR 0.88 [0.80, 0.96]
Recurrence after autotransfusion vs. none in hepatectomy	2	Very Low	OR 0.28 [0.04, 2.18]
OS after autotransfusion vs. none in hepatectomy	3	Very Low	HR 0.71 [0.58, 0.88]

## Data Availability

All data contributing to the analysis of this manuscript are available from the corresponding author upon reasonable request.

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
