# Peer review of "Effect of Autotransfusion in HCC Surgery on Survival and Recurrence: A Systematic Review and Meta-Analysis"

_cancers, 2022, doi:10.3390/cancers14194837_

Round 1
Reviewer 1 Report
I think this is a well written paper on the effects of autotransfusion in HCC surgery, a topic that is of significant interest to the reader, a topic of much controversy and discussion. The rationale is sound, your methods are well described and limitations are recognized appropriately. It is important to note that the pt who got auto transfusion in the hepatectomy group has a larger proportion of pts with CP-A disease with a significant p-value, which by itself explains better survival. This needs to be mentioned in the discussion section.
Author Response
We thank you for taking your valuable time to review our manuscript titled “Effects of autotransfusion in HCC surgery on survival and recurrence, a systematic review and meta-analysis”. We appreciate all the comments provided and hope we answered them to the satisfaction of the reviewers and editors. We feel that the comments have led to improvements of our manuscript.
Please find a point-by-point response to your remarks below.
Reviewer 1:
I think this is a well written paper on the effects of autotransfusion in HCC surgery, a topic that is of significant interest to the reader, a topic of much controversy and discussion. The rationale is sound, your methods are well described and limitations are recognized appropriately. It is important to note that the pt who got auto transfusion in the hepatectomy group has a larger proportion of pts with CP-A disease with a significant p-value, which by itself explains better survival. This needs to be mentioned in the discussion section.
Answer: We thank the reviewer for the positive assessment and the remark concerning Child Pugh classification. We have added this point in the discussion section (lines 211 – 213).
Reviewer 2 Report
The authors performed meta analysis on autotransfusion in HCC surgery. They analysed studies for liver transplantation as well as for hemihepatectomy. Interestingly studies regarding hemihepatectomy used autotransfusion but studies for transplantation IBS, so comparison is difficult. Furthermore, the authors could show at least a non-inferiority of autotransfusion/IBS in regards to overall survival and disease free survival after surgery. The analysis is important for future studies in time of blood shortage.
Comments:
- the manuscript is well written and addresses an important subject and the need for future research in this field
- only 15 studies could be selected and included in the meta analysis. Furthermore, due to different transfusion techniques (autotransfusion vs. IBS) hemihepatectomy and transplantation had to be analysed separately. This reduced included patients and the strength of the analysis is somewhat low. This should be stressed more as a big limitation of the study.
- Figure 1 shows the exclusion of 93 studies. Please state more in detail why those studies were excluded. As the total amount of studies included in the analysis is fairly low, this explanation is important.
- headline of table 4 needs to be formated
- table 5 missing a headline
Author Response
We thank you for taking your valuable time to review our manuscript titled “Effects of autotransfusion in HCC surgery on survival and recurrence, a systematic review and meta-analysis”. We appreciate all the comments provided and hope we answered them to the satisfaction of the reviewers and editors. We feel that the comments have led to improvements of our manuscript.
Please find a point-by-point response to your remarks below.
Reviewer 2:
The authors performed meta analysis on autotransfusion in HCC surgery. They analysed studies for liver transplantation as well as for hemihepatectomy. Interestingly studies regarding hemihepatectomy used autotransfusion but studies for transplantation IBS, so comparison is difficult. Furthermore, the authors could show at least a non-inferiority of autotransfusion/IBS in regards to overall survival and disease free survival after surgery. The analysis is important for future studies in time of blood shortage.
Comments:
- the manuscript is well written and addresses an important subject and the need for future research in this field
Answer: We thank the reviewer for the positive feedback and agree that more research is needed on the subject.
- only 15 studies could be selected and included in the meta analysis. Furthermore, due to different transfusion techniques (autotransfusion vs. IBS) hemihepatectomy and transplantation had to be analysed separately. This reduced included patients and the strength of the analysis is somewhat low. This should be stressed more as a big limitation of the study.
Answer: We thank the reviewer for the remark. We have added this point in the discussion section as a limitation and underlined that it influences the certainty of evidence. (lines 220 – 224).
- Figure 1 shows the exclusion of 93 studies. Please state more in detail why those studies were excluded. As the total amount of studies included in the analysis is fairly low, this explanation is important.
Answer: Thank you for your comment. We have updated figure 1 to include the reasons for exclusions. Mostly records were excluded because they did not contain the intervention of interest.
- headline of table 4 needs to be formatted
Answer: Thank you for your remark. We have formatted the headline of table 4.
- table 5 missing a headline
Answer: Thank you for your remark. We have edited the table to include a headline.